# Rapid Personalization of PDE-Based Tumor Growth using a Differentiable Forward Model

**Jonas Weidner**[1,2]                                                J.WEIDNER@TUM.DE
**Ivan Ezhov**[1,2]                                                   IVAN.EZHOV@TUM.DE
[1] *Dept. of Computer Science, Technical University of Munich, Germany*
[2] *Center for Translational Cancer Research, Technical University of Munich, Germany*

**Michal Balcerak**[3]                                               MICHAL.BALCERAK@UZH.CH
**Björn Menze**[3]                                                   BJOERN.MENZE@UZH.CH
[3] *Dept. of Quantitative Biomedicine, University of Zurich, Switzerland*

**Benedikt Wiestler**[2,4]                                           B.WIESTLER@TUM.DE
[4] *Dept. of Neuroradiology, School of Medicine and Health, Technical University of Munich, Germany*

**Editors:** Under Review for MIDL 2024

## Abstract

Partial differential equation (PDE) based brain tumor growth models have the potential to personalize glioma therapy. However, calibrating these models to individual patients is computationally expensive using traditional optimization techniques. In this work, we propose an approach leveraging the differentiability of deep learning (DL) based PDE forward solvers to efficiently calibrate the tumor models. Through gradient-based optimization with respect to the input tumor parameters, we iteratively minimize the loss between the predicted and actual tumor distribution in the patient's MRI scans. We evaluate our method on a cohort of nine glioma patients and demonstrate a dramatic reduction in the time to solve the inverse problem from hours, using typically employed evolutionary sampling or Monte Carlo methods, to minutes while achieving comparable modeling results.

**Keywords:** gradient-based optimization, PDE-based growth models, tumor modeling

## 1. Introduction

Despite advances in understanding the complex tumor biology and extensive clinical trials (Weller et al., 2021), the treatment of glioblastoma, the most common and most malignant form of primary brain tumors, remains a formidable challenge due to the tumor's tendency to spread into surrounding brain tissue. Radiotherapy is a cornerstone of treatment aimed at balancing effective tumor irradiation with minimal damage to surrounding healthy brain tissue. Current planning primarily uses MR and CT imaging to define a clinical target volume, which includes a uniform margin around the tumor to accommodate diffuse spread but lacks patient-specific customization (Niyazi et al., 2023).

Tumor modeling can help by providing a personalized and dynamic understanding of tumor growth and spread, enabling the design of targeted and more effective treatment strategies. Sampling-based methods like Markov Chain Monte Carlo offer high-precision modeling (Lipková et al., 2019). They require many costly forward runs, resulting in a typical runtime of multiple hours to days for the inverse fitting problem.

Deep learning methods, on the other hand, can directly predict tumor growth parameters without iterative optimization, finishing in minutes (Ezhov et al., 2023; Pati et al., 2021).

However, these DL approaches are yet far from optimal due to their inherent challenge to generalize reliably to unseen cases. In an attempt to improve the performance of such methods, we propose a technique combining the speed advantage of learnable methods with the adaptability of gradient-based optimization. We learn the well-posed forward simulation with a DL model and optimize its inputs to solve the inverse problem.

## 2. Methods

We aim to fit a PDE-based biophysical brain tumor model with unknown tumor parameters $\theta$ to a patient's MR image[1].

**Forward Tumor Solver:** Tumor growth encompasses two fundamental processes: proliferation and diffusion. Thus, the progression of the tumor cell concentration $c$ can be described by a PDE with diffusion coefficient $D$ and growth rate $\rho$ as free parameters $\theta = \{\rho, D\}$[2]: $\frac{\partial c}{\partial t} = \nabla \cdot (D\nabla c) + \rho c(1 - c)$. In addition to the tumor parameters, the model inputs the patient-specific tissue anatomy[3] $T$ and outputs a prediction $\hat{c}$ for the tumor concentration $c$. A detailed description is provided in (Ezhov et al., 2023). We replace the typically used non-differentiable numerical solver with a neural network $f$, parametrized by weights $\phi$, to predict the tumor concentration $\hat{c} = f_\phi(\theta, T)$. Here, we adapted the learnable forward solver from (Ezhov et al., 2021) and trained it on 10,000 simulations within a brain atlas (Rohlfing et al., 2010) created by a high-precision numerical solver (Lipková et al., 2019).

**Inverse Parameter Optimization:** To efficiently solve the inverse problem, we employ gradient-based optimization (Figure 1). Therefore, we start with an initial forward run and backpropagate the loss in predicted tumor concentration with respect to the tumor parameters $\nabla_\theta \hat{c} = \nabla_\theta f_\phi(T, \theta)$ through the network. It is important to note that we do not derivate with respect to the (fixed) weights $\phi$ but to the input parameters $\theta$. Utilizing this gradient, we can iteratively update $\theta$ with a gradient-based optimization step $S$ as: $\theta_{\text{New}} = S(\theta, \nabla_\theta f_\phi(T, \theta))$. For that, we used the Adam optimizer (Kingma and Ba, 2014). By fixing the network's weights ($\phi$) when we solve the inverse problem, we aim to adhere to physical modeling constraints in our solution.

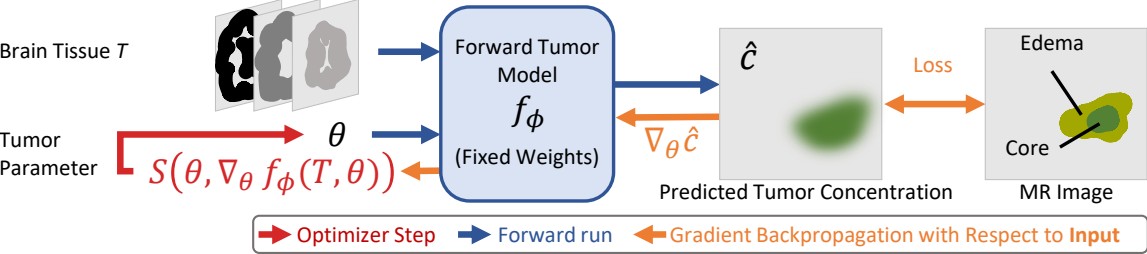

Figure 1: Personalizing tumor models to an individual patient. After initialization with tumor parameters and tissue, the forward tumor model predicts a tumor concentration. Then, the tumor parameters are optimized iteratively using backpropagated gradients of the loss between the prediction and the MRI.

---

1. Tumor segmentation of edema and contrast-enhancing tumor.
2. The end time is normalized and the tumor origin is fixed at the center of mass of the tumor segmentation.
3. The segmented white and gray matter and CSF masks.

## 3. Experiments and Results

We first evaluate our method on synthetic cases, where we achieve a Dice score of $0.92 \pm 0.01$ for edema and $0.89 \pm 0.02$ for the tumor core. Next, we validate our method in nine glioblastoma patients by comparing the predicted tumor concentration with the observed tumor segmentations. We assume a typical tumor concentration threshold of 0.25 for edema and 0.675 for the tumor core to threshold tumor concentration into segmentations (Ezhov et al., 2023). We find that our method performs remarkably similar to the standard evolutionary sampling[4] at a runtime comparable to inverse DL methods, which we clearly outperform in accuracy (Table 1). An example patient is shown in Figure 2.

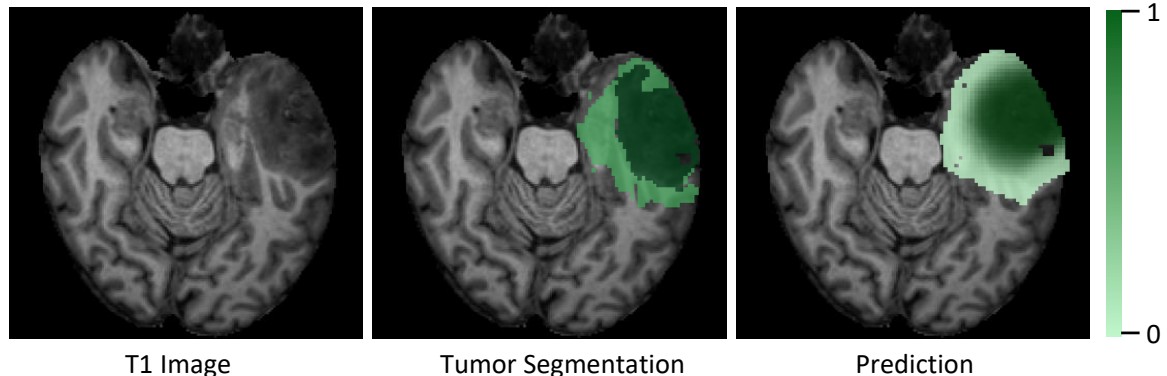

T1 Image       Tumor Segmentation       Prediction

Figure 2: Demonstration of our method on one patient. The images show a patient's MR scan, the segmentations of the contrast-enhancing and edema region, and the optimized network prediction of the tumor concentration $\hat{c}$.

Table 1: Dice score in contrast-enhancing tumor (CET) and edema as well as the runtimes in nine glioblastoma patients.

| Method | Dice CET | Dice Edema | Runtime [min] |
|---|---|---|---|
| Ours | **0.75** $\pm$ 0.02 | 0.65 $\pm$ 0.03 | 1.2 $\pm$ 0.1 |
| Evolutionary Sampling | 0.70 $\pm$ 0.03 | **0.70** $\pm$ 0.02 | 978 $\pm$ 102 |
| One-shot DL (Ezhov et al., 2023) | 0.34 $\pm$ 0.09 | 0.55 $\pm$ 0.04 | < 1 min |

**Conclusion:** Our method, leveraging the differentiability of a deep learning solver for gradient-based personalization of a growth model, shows promising results on real patient cases. It combines the speed of DL methods with the adaptability of gradient-based optimization. These results warrant future studies into more complex growth models and network architectures.

## Acknowledgments

This work was supported in part by the DFG (grant# 504320104) and the NIH (grant# R01CA269948).

---

4. We used CMA-ES (Hansen and Ostermeier, 2001) with the forward solver of (Lipková et al., 2019).

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
