# OpenReview forum: "Rapid Personalization of PDE-Based Tumor Growth using a Differentiable Forward Model"
_MIDL.io/2024/Short_Papers — MIDL 2024 Short Papers_

### Official Review · Reviewer_ossr · 2024-04-24

**Confidence:** 5
**Final Rating:** 5

**Review:**

This paper formulates the tumor progression as a diffusion model and tumor concentration prediction in the patient’s anatomy and applies a learnable solver based on extensive simulation data. Incorporating the neural networks as the learnable solver sounds novel. The method is well-designed and detailed. The paper demonstrates promising results (significant performance increases compared to a recent one-shot deep learning method and substantial runtime reduction compared to evolutionary sampling).

---

### Decision · Program_Chairs · 2024-04-26

Accept